# Evaluation of Quality of Life in Adult Celiac Patients Living in Lithuania and Their Compliance with a Gluten-Free Diet: A Pilot Study

**DOI:** 10.3390/medicina61071278

**Published:** 2025-07-16

**Authors:** Yeliz Serin, Jurgita Andruškienė, Anil K. Verma, Monika Śmiełowska, Nerijus Dzingelevičius, Albinas Vilčiauskis, Rita Vaičekauskaitė, Vijolė Bradauskienė, Bogusław Buszewski, Reda Dzingelevičienė

**Affiliations:** 1Department of Nutrition and Dietetics, Faculty of Health Sciences, Cukurova University, Adana 01330, Türkiye; 2Faculty of Health Science, Klaipeda University, H. Manto Str. 84, 92294 Klaipeda, Lithuania; jurgita.andruskiene@ku.lt (J.A.); monika.barbara.smielowska@ku.lt (M.Ś.); rita.vaicekauskaite@ku.lt (R.V.); bbusz@umk.pl (B.B.); reda.dzingeleviciene@ku.lt (R.D.); 3Department of Medicine, Farncombe Family Digestive Health Research Institute, McMaster University, Hamilton, ON L8S 4K1, Canada; vermaa61@mcmaster.ca; 4Celiac Disease Research Laboratory, Polytechnic University of Marche, 60121 Ancona, Italy; 5Kuyavian-Pomeranian Science and Technology Center of Prof. Jan Czochralski in Toruń, Krasińskiego Str. 4, 87-100 Toruń, Poland; 6Faculty of Technologies, Klaipėdos Valstybinė Kolegija/Higher Education Institution, Bijūnų Str. 10, 91223 Klaipeda, Lithuania; nerijus.dzingelevicius@ku.lt (N.D.); v.bradauskiene@kvk.lt (V.B.); 7Academy of Medicine, Faculty of Medicine, Lithuanian University of Health Sciences, A. Mickevičiaus Str. 9, 44307 Kaunas, Lithuania; albinasvil@gmail.com

**Keywords:** celiac disease, quality of life, gluten-free diet

## Abstract

*Background and Objectives*: Celiac disease (CD) is a common gluten-related disorder associated with significantly worsened quality of life. The aim of this pilot study was to evaluate the quality of life of adult celiac patients living in Lithuania and their compliance with a gluten-free diet. *Materials and Methods*: This cross-sectional study was conducted on individuals aged 18 to 75 years diagnosed with CD and residing within the borders of Lithuania. This pilot study involved 73 participants, comprising 68 females and 5 males. The CD Specific Quality of Life Scale (CD-QOL) consisted of 20 items across four sub-dimensions. Responses to scale items were graded with a score ranging from 1 (not at all) to 5 (very much). The total score obtained from the scale can range up to 100, with a score < 40 classified as poor, 40–50 as moderate, and > 50 as good quality of life. Additionally, ten questions related to gluten-free diet-related quality of life were used. *Results*: The mean age of diagnosis for females (32.6 ± 11.7) was higher than that for males (22.0 ± 12.1), *p* < 0.05. The mean self-reported BMI for males (25.8 ± 4.5) was higher than that for females (22.3 ± 5.2), *p* < 0.05. The mean quality-of-life score (66.4 ± 12.5) was significantly higher in the good quality-of-life group compared to the poor group (33.7 ± 3.9), *p* < 0.001. Half of the respondents (50.7%) reported that gluten-free products are expensive, and 45.2% confirmed difficulties in dining out on a gluten-free diet. *Conclusions*: The results of this pilot study indicate that CD is associated with a worsened quality of life and that compliance with a gluten-free diet is primarily influenced by economic factors, such as the high cost of the diet.

## 1. Introduction

Celiac disease (CD) is an autoimmune condition of the small intestine that develops in genetically predisposed individuals as a permanent intolerance to gluten in wheat and other gluten-like grain proteins in grains such as barley, rye, and oats. Gluten-sensitive enteropathy (GSE) is also called celiac sprue. It develops through autoimmune mechanisms [1].

CD is one of the most common gluten intolerance diseases that is often underdiagnosed [2]. The prevalence of CD in developed countries is estimated to be approximately 0.5–1%. This rate is increasing worldwide due to the trigger effect of environmental factors [3].

The only treatment for CD is a lifelong gluten-free diet [4]. Patients who follow a gluten-free diet positively increase their daily energy intake and body mass index (BMI), and their complications associated with CD decrease. However, it is very difficult for many patients to maintain a gluten-free diet for a lifetime. [5]. In one study, it was determined that the rate of individuals adapting to a gluten-free diet varied between 42% and 91% [6].

Some nutritional deficiencies may develop in celiac patients due to a long-term gluten-free diet. In particular, deficiencies in B vitamins and fiber intake may occur due to the elimination of carbohydrate and grain sources [7]. In addition, due to damage to the intestinal mucosa, there is often a decrease in levels of iron, vitamin B12, folic acid, and vitamin D [8].

According to the study by Zarkadas et al. (2013) on adult celiac patients, it was determined that inadequate food label information and eating outside the home were the most common problems during adherence to a gluten-free diet, and that patients developed feelings of “deprivation and isolation” due to the diet [3]. In a study conducted in the United States, celiac patients evaluated their general health status as “poor” [9].

There are limited population-based large-scale studies on adult celiac patients worldwide [10,11,12]. Diet-related nutritional deficiencies frequently occur in celiac patients. In addition, CD affects the quality of life of individuals in social, psychological, and economic terms. No study has been found that evaluates quality of life and compliance with a gluten-free diet in celiac patients living in Lithuania. The aim of this study was to evaluate quality of life and adherence to a gluten-free diet in adult celiac patients living in Lithuania.

## 2. Materials and Methods

### 2.1. Type of Research, Place, and Time

This study was cross-sectional and descriptive, and was planned to be conducted on individuals aged 18–75 years who had been diagnosed with CD and reside within the borders of Lithuania.

### 2.2. Samples Selection of the Research

This study was conducted with individuals diagnosed with CD in Lithuania who were registered in various celiac-related social support networks (Facebook, Instagram, email, and WhatsApp) and agreed to participate in the study. Patients aged between 18–75 years, diagnosed with CD with biopsy confirmation, able to make food choices on their own, who had been following a GFD for at least 6 months, who completed the online questionnare on their own, and who had no communication issues, were included in this study.

### 2.3. Calculation of Sample Size

The sample size calculation for this study was determined by taking as a reference the article “Coeliac disease: the association between quality of life and social support network participation” by Lee et al. (2016), with an independent sample *t* test, a 0.68 effect size (Cohen’s d), a 0.05 margin of error, and a 0.80 power; the minimum sample number to be included in this study was 35 for each group and a total of at least 70 people [13]. The sample size estimation for this study was made with the G.Power 3.1.9.4 program [14].

### 2.4. Research Data Collection

The survey form was created online via Google Docs. Google Docs is a word processor that is part of a free, web-based office software suite offered by Google under its Google Drive Service [15]. In order to avoid data duplication, only one response entry was allowed from each email address. This ensured that participants filled in their responses without skipping any questions and without data duplication by real people. The online survey form was delivered to participants free of charge via links on social media applications (Facebook, Instagram, email, and WhatsApp) and in groups created on these platforms. According to the Helsinki Declaration, individuals participating in this study were informed about the purpose and scope of this research. Ethics Committee approval was obtained for this research from the Çukurova University Faculty of Medicine Research Ethics Committee with decision number 39, dated 6 December 2024 [16]. Detailed information about this study and researchers was provided on the first page of the online survey form, and online consent was obtained from individuals indicating that they voluntarily agreed to participate in this study. The questionnaire (Appendix A) and related explanations were first prepared in English and then translated from English to Lithuanian by people with proficiency in both languages. In the final stage, the results were translated from Lithuanian to English. The questionnaire form used in this study consisted of three sections. The first section of the questionnaire included sociodemographic characteristics, the second section included the CD-related quality-of-life scale, and the third section included the gluten-free diet compliance questionnaire.

### 2.5. Celiac Disease Specific Quality of Life Scale (CD-QOL)

This scale form consisted of 20 items and four sub-dimensions (Limitations, Dysphoria, Health Concerns, and Inadequate Treatment). Responses to scale items were graded with a score ranging from 1 (not at all) to 5 (very much). The total score that could be obtained from the scale was 100, and a scale score < 40 was classified as poor, 40–50 as moderate, and >50 as good quality of life [13,17]. The Cronbach’s alpha coefficient of the scale was 0.73 [17].

### 2.6. Gluten-Free Diet-Related Quality-of-Life Questionnaire

The quality-of-life questionnaire form consisting of 10 questions related to gluten-free diet was created by compiling questions from previously published studies [9,18,19]. The effects of a gluten-free diet on social life were evaluated by 10 questions in the survey on a scale of 1 (not at all) to 5 (a lot). These questions were incorporated to learn more about specific aspects of following a gluten-free diet that might contribute to quality of life that were not included on the CD-QOL.

### 2.7. Statistical Analyses

In order to conduct the statistical analyses, the program known as SPSS 22.0 was utilized. To check for normality, dependent and independent variables were analyzed using Kolmogorov–Smirnov and Shapiro–Wilk tests and visual tools like probability plots and histograms. Parametric tests were used for quantitative variables with normally distributed distributions (the independent group *t* test and ANOVA), while non-parametric tests (the Mann–Whitney U test and the Kruskal–Wallis test) were employed for variables with non-normal distributions. For variables with normally distributed data, descriptive analyses were conducted using the mean and standard deviation. For variables with non-normally distributed data, the median and interquartile range were used. For parameters with normally distributed variables, we used the Pearson test; for those with non-normally distributed variables, we used the Spearman test to determine the correlation coefficients. To indicate a statistically significant outcome, a *p* value less than 0.05 was used [20].

## 3. Results

This study involved a total of 73 participations, comprising 68 females and 5 males. The educational attainment of study participants predominantly comprised university. Most of them settled in the city (Table 1).

The mean age at diagnosis in females was higher than it was in males. The BMI of males was higher than that of females (*p* < 0.05). Quality-of-life scores of females were lower than those of males were (*p* > 0.05) (Table 2).

The mean age and age at diagnosis of the medium group were higher than those of the poor and good groups (*p* > 0.05). The quality-of-life score of the good group was higher than those of the poor and median groups (*p* < 0.05, Table 3).

The two items that appeared most problematic for participants were related to the cost of gluten-free products (50.7%) and difficulty dining out (45.2%). In comparison, participants answered “not at all” to items regarding whether following a gluten-free diet was embarrassing (35.6%),) or limited their social life (43.8%), and if friends/family did not understand their need to follow the diet (60.3%). Thirty-eight percent of participants reported “slightly” to the statement that a gluten-free diet was difficult (38.4%) (Table 4).

## 4. Discussion

The educational level of patients may influence their understanding and perception of CD processes. In addition, where CD patients live may be influenced by differences in the availability and accessibility of GF products between urban and rural areas [21,22]. Educational level can increase patients’ knowledge regarding their disease, leading to improvements in their health [23]. The educational attainment of study participants predominantly comprised university, and most of them settled in the city (Table 1).

There is a female predominance over males in diagnosed CD (2:1 to 10:1) [24,25,26]. In this study, the diagnosed age of women was higher than that of men (Table 2; *p* < 0.05). CD can affect women’s health by decreasing bone mineral density and fertility, increasing the risk of concurrent autoimmune disease, and negatively impacting quality of life [26].

For many years, CD has been considered an absorption disorder associated with classical CD, generally considered in individuals presenting to various clinics with symptoms of growth and developmental delay in children, weight loss in children, or failure to gain body weight [27]. However, with the recent widespread use of serological tests, some studies have shown that patients’ BMI values are higher than the normal range (18.5–24.9 kg/m^2^) at the beginning or later stages of celiac diagnosis [28,29,30]. Therefore, it should not be forgotten that individuals with normal or higher BMI values may also be at risk of CD [31]. In a study comparing the BMI values of celiac patients with those of a healthy population, the average BMI values of male patients were determined to be 21.6 ± 2.9 kg/m^2^, and the average BMI values of female patients were determined to be 23.1 ± 4.0 kg/m^2^ [32]. Similarly, in this study, women’s BMI values were found to be normal (22.3 ± 5.2 kg/m^2^) and men’s BMI values were found to be overweight (25.8 ± 4.5 kg/m^2^) (Table 2).

To date, many different types of questionnaires, such as the Short Form Health Survey 36—SF36 [12], the Short Form Health Survey 12—SF12 [33], the Psychological General Well-Being Index—PGWB [34], the World Health Organisation Quality of Life—WHOQOL assessment [35], Health Related Quality of Life—HRQOL assessments [36]., and the Hospital Anxiety and Depression Scale—HADS [37], have been used to assess the quality of life of celiac patients. However, these questionnaires are not specific to any disease and are generally used to determine the depression and/or anxiety states of individuals and the relationship between their general health status and quality of life [38]. In this study, a CD-specific health-related quality-of-life questionnaire developed by Dorn et al. was used (Table 3) [17].

Currently, the only treatment for CD is to follow a lifelong gluten-free diet. Although this provides an advantage for individuals in terms of undertaking their own treatment, it is not easy to change their usual diet, especially for adult celiac patients [39]. Studies have shown that health-related quality of life in CD is associated with many factors, such as gender, age, marital status, education level, disease duration, gastrointestinal system findings, compliance with a gluten-free diet, comorbid disease status, and age at diagnosis [39,40,41,42]. In this study, the quality of life of celiac patients was associated with age, age at diagnosis, and BMI (Table 3).

It has been assumed that increases in health-related quality-of-life scores after the histologic recovery of small intestinal mucosa occur within 6 to 12 months after starting a gluten-free diet, simultaneously with clinical remission [43]. The recovery of the intestinal tract leads to increases in positive health-related quality-of-life scores of celiac patients [35]. On the other hand, celiac patients who suffer persistent symptoms despite adherence to a gluten-free diet are at a greater risk of reduced health-related quality-of-life scores [44]. Extraintestinal manifestations of celiac disease, such as autoimmune thyroid disease, dermatitis herpetiformis, type 1 diabetes, irritable bowel syndrome, unexplained neuropsychiatric disorders, neurological disorders (attention/memory impairment, peripheral neuropathy, and others), and nonalcoholic fatty liver disease, may influence health-related quality of life [45,46].

In addition to cognitive, emotional, and socio-cultural interactions, factors such as whether individuals are regularly followed by health professionals or are members of a celiac support group also affect individuals’ compliance with a gluten-free diet [6]. On the other hand, it is difficult to maintain a gluten-free diet throughout life due to the high prices of gluten-free foods, the low quality of gluten-free products in terms of taste and texture, limited access to food by patients, unpredictable gluten contamination, age at diagnosis, the duration of a gluten-free diet, treatment burden, a lack of knowledge and information about celiac disease and the gluten-free diet, psychological factors in celiac patients, the negative effects of a gluten-free diet, and different cultural practices in which food is served, which can cause significant social problems [3,47,48,49,50,51].

In one study, individuals complained most about food labels not being descriptive enough, preparing gluten-free foods too often, other people making the individual feel upset about their illness, people thinking that a small amount of gluten would not cause harm, and limited food options outside the home [3]. In another study, individuals on a gluten-free diet stated that they felt isolated from social life [52]. In a study conducted in the UK, individuals mostly complained about having to carry gluten-free food with them at all times during traveling, and sometimes about the inadequacy of product packaging information and not being able to attend social invitations [53]. In this study, the two items that appeared most problematic for participants were related to the cost of gluten-free products (50.7%) and difficulty dining out (45.2%). In comparison, participants answered “not at all” to items regarding whether following a gluten-free diet was embarrassing (35.6%) or limited their social life (43.8%), and if friends/family did not understand their need to follow the diet (60.3%). Thirty-eight percent of participants reported “slightly” to the statement that following a gluten-free diet was difficult (38.4%) (Table 4).

The strength of this study is that, to our knowledge, it is the first study to evaluate the quality of life and compliance with a gluten-free diet in individuals with CD in Lithuania. Online questionnaires have some advantages, as they are quick and easy to answer, and it is easier to analyze and interpret the data obtained.

The main limitation of this study was its very small sample size, which is a common feature of pilot studies. On the other hand, this study was conducted only in one city (Klaipeda, with a population size of 159,396) in Lithuania. This might have affected the sample size of this study. Further investigations are needed to obtain a deeper understanding of the impact of CD on the quality of life in the general population. The scales that were used in this study did not have validated Lithuanian translations. Data collection for this study was conducted using an online survey method. Compared to online surveys, face-to-face surveys are a more controllable method in many respects, as there is mutual contact and interaction between the researcher and the respondent until the survey ends. We did not collect any data about extraintestinal manifestions and how celiac disease patients received nutritional education and information about gluten-free food.

## 5. Conclusions

The results of this pilot study show that celiac disease is associated with a worsened quality of life. However, a greater number of detailed studies are needed to investigate the quality-of-life areas that are mostly affected in the general population of Lithuania. Compliance with a gluten-free diet was affected by the high cost of a gluten-free diet and difficulty dining out on a gluten-free diet.

## Figures and Tables

**Table 1 medicina-61-01278-t001:** Sociodemographıc data of participants.

Education	N	%
Secondary	5	6.8
College	7	9.6
University	61	83.6
Place of settlement		
City	58	79.5
Settlement	4	5.5
Town	6	8.2
Village	5	6.8
Gender		
Female	68	93.2
Male	5	6.8

**Table 2 medicina-61-01278-t002:** Evaluation of age at diagnosis, body mass index, and quality of life by gender.

	Female (n = 68)		Male (n = 5)		*p* Value
	Mean ± SD	Median (IQR)	Mean ± SD	Median (IQR)	
Age at diagnosis	32.6 ± 11.7	33.0 (13.0)	22.0 ± 12.1	28.0 (18)	0.04 *
Body mass index	22.3 ± 5.2	21.2 (4.5)	25.8 ± 4.5	24.7 (8.4)	0.04 *
Quality-of-life score	49.8 ± 18.0	45.5 (24)	56.8 ± 14.7	60.0 (28.0)	0.31

Mann Whitney U test, * *p* < 0.05.

**Table 3 medicina-61-01278-t003:** Comparison of quality-of-life scores according to some parameters.

	Quality of Life						
Parameters	Poor (n = 30)		Medium (n = 9)		Good (n = 34)		*p* Value *
	Mean ± SD	Median (IQR)	Mean ± SD	Median (IQR)	Mean ± SD	Median (IQR)	
Age	34.4 ± 8.9	35 (9.5)	37.1 ± 8.6	35.0 (12.5)	36.9 ± 10.7	34.0 (13.3)	0.09
Age at diagnosis	31.1 ± 10.2	33.0 (11.0)	34.8 ± 9.5	32.0 (11.5)	31.7 ± 13.9	31.5 (15.5)	0.74
BMI	21.0 ± 2.1	20.6 (3.3)	23.4 ± 6.8	20.9 (7.5)	23.7 ± 6.3	22.5 (5.5)	0.83
Quality of life	33.7 ± 3.9	34.5 (6.3)	44.6 ± 2.2	45.0 (3.5)	66.4 ± 12.5	63.5 (19.3)	0.001

Kruskal–Wallis Test, * *p* < 0.05.

**Table 4 medicina-61-01278-t004:** Gluten-free diet specific quality-of-life survey responses.

	Not at All		Slightly		Moderately		Quite a Bit		A Great Deal	
	N	%	N	%	N	%	N	%	N	%
Following the gluten-free diet is difficult	14	19.2	28	38.4	17	23.3	11	15.1	3	4.1
Gluten-free products are expensive	-	-	15	20.5	4	5.5	17	23.3	37	50.7
A gluten-free diet is socially isolating	10	13.7	30	41.1	8	11.0	14	19.2	11	15.1
Gluten-free food is tasteless	29	39.7	15	20.5	17	23.3	9	12.3	3	4.1
Friends/family don’t understand my need to follow the diet	44	60.3	17	23.3	2	2.7	3	4.1	7	9.6
Following a gluten-free diet is embarrassing	26	35.6	22	30.1	10	13.7	9	12.3	6	8.2
Difficulty in finding gluten-free foods in grocery stores	5	6.8	25	34.2	8	11.0	21	28.8	14	19.2
Difficulty in dining out on the gluten-free diet	1	1.4	17	23.3	10	13.7	12	16.4	33	45.2
Because of the gluten-free diet I choose not to dine out	20	27.4	17	23.3	9	12.3	14	19.2	13	17.8
Because of the diet I limit my social life	32	43.8	22	30.1	8	11.0	6	8.2	5	6.8

## Data Availability

Datasets used and/or analyzed during the current study are available from the corresponding author upon reasonable request.

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
