# Peer review of "Evaluation of Quality of Life in Adult Celiac Patients Living in Lithuania and Their Compliance with a Gluten-Free Diet: A Pilot Study"

_medicina, 2025, doi:10.3390/medicina61071278_

Round 1
Reviewer 1 Report
Comments and Suggestions for Authors
The authors have explored the quality of life and adherence to a gluten-free diet in adult celiac patients in Lithuania. Following are my comments/questions:
- The sample size (n=73) is small and highly skewed towards females (93%), which can limit generalizability. Can the authors clarify recruitment strategies and discuss how gender imbalance may affect findings?
- The analysis is univariate, using Mann–Whitney U and Kruskal–Wallis tests, without controlling for confounders such as age or BMI. Why were multivariate models (e.g., linear or logistic regression) not applied to adjust for potential confounding factors?
Author Response
Reviewer 1
The authors have explored the quality of life and adherence to a gluten-free diet in adult celiac patients in Lithuania. Following are my comments/questions:
- The sample size (n=73) is small and highly skewed towards females (93%), which can limit generalizability. Can the authors clarify recruitment strategies and discuss how gender imbalance may affect findings?
Thanks for this comment. Studies have confirmed that females are more affected with celiac disease in comparison to males ( 2:1 to 10:1). In our study as well, the sampling distribution is consistent with the literature. This section was discussed in lines 180-183.
Additionally, we accept that the sample size was low in our study because the incidence /prevalence of celiac disease is 0.8-1.0 % in some European countries. On the other hand, there is no specific epidemiological data for Lithuania. The dataset consists of only one city (Klaipeda) in Lithuania. Its population is 159,396. The population size affects the sample size of the study. We have this information in lines 257-260
- The analysis is univariate, using Mann–Whitney U and Kruskal–Wallis tests, without controlling for confounders such as age or BMI. Why were multivariate models (e.g., linear or logistic regression) not applied to adjust for potential confounding factors?
Thanks a lot for raising this. The logistic or linear regression models could not be applied due to the low sample size and extreme (outlier) values ( Reference at below 1 and 2).
The exrteme values:
(BMI min :16.4 kg/m2- max :53.3 kg/m2; quality of life scores (min 26.0-max:95 );diagnosed age ( min :1.0 max : 56.0)
References
1)Bujang MA, Sa'at N, Sidik TMITAB, Joo LC. Sample Size Guidelines for Logistic Regression from Observational Studies with Large Population: Emphasis on the Accuracy Between Statistics and Parameters Based on Real-Life Clinical Data. Malays J Med Sci. 2018 Jul;25(4):122-130. doi: 10.21315/mjms2018.25.4.12. Epub 2018 Aug 30. PMID: 30914854; PMCID: PMC6422534.
2) Lukman AF, Mohammed S, Olaluwoye O, Farghali RA. Handling Multicollinearity and Outliers in Logistic Regression Using the Robust Kibria–Lukman Estimator. Axioms. 2025; 14(1):19. https://doi.org/10.3390/axioms14010019

Reviewer 2 Report
Comments and Suggestions for Authors
I appreciate the opportunity to review this interesting article. It is a cohort study evaluating the quality of life and adherence to a gluten-free diet in adult patients with celiac disease living in Lithuania. The introduction is adequate, citing scientific references relevant to the topic and justifying the study. The presentation of the results and the discussion and conclusions are adequate, with clear information and comparisons to relevant studies. Overall, I have only a few comments.
First, I recommend adding the survey administered to the study participants in both English and translated versions, perhaps as an appendix to the article.
Do the scales used in the study have validated translations? If these translations are unavailable, there should be a more in-depth explanation of how the quality of the translation was assessed for correct interpretation by the patients. If the translations are validated, the article in which they were validated must be cited. Otherwise, the study limitations should include the fact that the corresponding surveys are not validated in Lithuanian.
In the study limitations section, limitations resulting from using an online tool to conduct the survey should be further explored.
Author Response
Reviewer 2
- First, I recommend adding the survey administered to the study participants in both English and translated versions, perhaps as an appendix to the article.
Thanks for asking this. We have given it in the supplemantary material
2) Do the scales used in the study have validated translations? If these translations are unavailable, there should be a more in-depth explanation of how the quality of the translation was assessed for correct interpretation by the patients. If the translations are validated, the article in which they were validated must be cited. Otherwise, the study limitations should include the fact that the corresponding surveys are not validated in Lithuanian. In the study limitations section, limitations resulting from using an online tool to conduct the survey should be further explored.
We are happy that you have asked this. The quality of the translation was assessed for correct interpretation by the patients, is explained in lines 104-106. The questionnaire and related explanations were first prepared in English and then translated from English to Lithuanian by people with both language proficiency.
Celiac Disease Specific Quality of Life Scale (CD-QOL) was selected according to Cronbach's alpha value (validation and reliability coefficient) . This information was added to line 116-117.
Gluten-Free Diet-Related Quality of Life Questionnaire was created by compiling from previously published studies. This information was given in line 119-120.
The scales that used in the study have not validated translations. It emphaized in limitation section (lines 260-261)
When compare online survey and face to face interview method, there is advantages and disadvantages between them. Rapid developments in the information and technology age have made it necessary for researchers to question data collection methods from an innovative and contemporary perspective. Online questionnaire has some advantages owing to quick and easy to answer, easier to analyze and interpret the data obtained ( line 254-255) Compared to online surveys, face-to-face surveys are a controllable method in many respects, as there is mutual contact and interaction between the researcher and the respondent until the survey ends. İt emphazied in line 262-264

Reviewer 3 Report
Comments and Suggestions for Authors
In this pilot cross-sectional study, the authors aimed to evaluate the quality of life and adherence to a gluten-free diet in 73 adult (68 females and 5 males) celiac disease patients living in Lithuania. They used the CD Specific Quality of Life Scale (CD-QOL), consisting of 20 items across 4 sub-dimensions (Limitations, Dysphoria, Health Concerns and Inadequate Treatment)..
Responses to scale items were graded with a score ranging from 1 (not at all) to 5 (very much), with a total score ranging up to 100, with a score of <40 classified as poor, 40-50 as moderate, and >50 as good quality of life. An additional 10 questions related to gluten-free diet-related quality of life were used. They found that the mean quality of life score (66.4 ± 12.5) was significantly higher in the good quality of life group compared to the poor group (33.7 ± 3.9), p < 0.001. Half the respondents (50.7%) reported that gluten-free products are expensive, and 45.2% confirmed difficulties in dining out on a gluten-free diet.
They concluded that CD is associated with a worsened quality of life, and compliance with the gluten-free diet is mainly influenced by economic issues, such as the high cost of the diet. The study offers an original report from celiac disease cohort patients in Lithuania and addresses the important topic of quality of life and its correlation with gluten-free diet costs.
There are, however, some issues requiring further information that should be addressed:
1)-Gluten-free diet: the authors should describe how celiac disease patients received nutritional education and information about gluten-free food and dietary instruction to avoid gluten contamination and any issues related to the need for strict dietary adherence. This is a very important issue, as celiac disease patients' social life may be significantly affected by restrictions secondary to dietary needs. As a consequence, previous studies have reported unsatisfactory dietary adherence, particularly in younger patients, and the median age of enrolled patients might reflect this potential confounding factor;
2)-Another important point is the clinical features of celiac disease patients. Quality of life may be related not only to abdominal/digestive symptoms (which resolve after gluten-free diet introduction) but also to the presence of extraintestinal manifestations, not always resolved after gluten-free diet initiation. In particular, the authors should recall in the discussion the potential occurrence of extraintestinal immune-related symptoms, such as neurological disorders (attention/memory impairment, peripheral neuropathy, and others), as previously demonstrated (DOI:10.1080/003655202761020542; DOI:10.1053/j.gastro.2007.04.070.) as well as metabolic changes related to gluten-free diet such as the Increased risk of nonalcoholic fatty liver disease in patients on a gluten-free diet, beyond traditional metabolic factors.
All of these issues should also be investigated by a questionnaire to accurately assess the quality of life and the impact of a gluten-free diet.
Author Response
Reviewer 3
There are, however, some issues requiring further information that should be addressed:
1)-Gluten-free diet: the authors should describe how celiac disease patients received nutritional education and information about gluten-free food and dietary instruction to avoid gluten contamination and any issues related to the need for strict dietary adherence. This is a very important issue, as celiac disease patients' social life may be significantly affected by restrictions secondary to dietary needs. As a consequence, previous studies have reported unsatisfactory dietary adherence, particularly in younger patients, and the median age of enrolled patients might reflect this potential .
Thanks for bringing this to us. We have any data about how celiac disease patients received nutritional education and information about gluten-free food and dietary instruction to avoid gluten contamination, and any issues related to the need for strict dietary adherence. This part was added to the limitation section, lines 264-267. However, the factors that influence gluten-free diet adaptation were questioned by the Gluten-Free Diet-Related Quality of Life Questionnaire (Table 4). In addition, the factors that affect gluten-free diet adaptation were discussed in lines 209-236.
2)-Another important point is the clinical features of celiac disease patients. Quality of life may be related not only to abdominal/digestive symptoms (which resolve after gluten-free diet introduction) but also to the presence of extraintestinal manifestations, not always resolved after gluten-free diet initiation. In particular, the authors should recall in the discussion the potential occurrence of extraintestinal immune-related symptoms, such as neurological disorders (attention/memory impairment, peripheral neuropathy, and others), as previously demonstrated (DOI:10.1080/003655202761020542; DOI:10.1053/j.gastro.2007.04.070.) as well as metabolic changes related to gluten-free diet such as the Increased risk of nonalcoholic fatty liver disease in patients on a gluten-free diet, beyond traditional metabolic factors. All of these issues should also be investigated by a questionnaire to accurately assess the quality of life and the impact of a gluten-free diet.
We are happy that you have asked about it. This section is discussed in lines 216-226
It has been expected that the health-related quality of life scores increase after histologic recovery of small intestinal mucosa is assumed to occur within 6 to 12 months after starting a gluten-free diet, simultaneously with clinical remission. The recovery of the intestinal tract leads to an increase in positive health-related quality of life scores of celiac patients.On the other hand, celiac patients who suffer persistent symptoms despite adherence to a gluten-free diet are at a greater risk of reduced health-related quality of life scores. Extraintestinal manifestations of celiac disease such as autoimmune thyroid disease, dermatitis herpetiformis and type 1 diabetes, irritable bowel syndrome, unexplained neuropsychiatric disorders, neurological disorders (attention/memory impairment, peripheral neuropathy, and others), and nonalcoholic fatty liver disease may influence health relted quality of life .
Casellas, F.; Rodrigo, L.; Vivancos, J.L.; Riestra, S.; Pantiga, C.; Baudet, J.-S.; Junquera, F.; Diví, V.P.; Abadia, C.; Papo, M. Factors that impact health-related quality of life in adults with celiac disease: a multicenter study. World journal of gastroenterology: WJG 2008, 14, 46.
Wahab, P.J.; Meijer, J.W.; Mulder, C.J. Histologic follow-up of people with celiac disease on a gluten-free diet: slow and incomplete recovery. American journal of clinical pathology 2002, 118, 459-463.
Harnett, J.E.; Myers, S.P. Quality of life in people with ongoing symptoms of coeliac disease despite adherence to a strict gluten-free diet. Scientific Reports 2020, 10, 1144.
Pinto-Sánchez, M.I.; Bercik, P.; Verdu, E.F.; Bai, J.C. Extraintestinal manifestations of celiac disease. Digestive Diseases 2015, 33, 147-154.
Volta, U.; Giorgio, R.D.; Petrolini, N.; Stanghellini, V.; Barbara, G.; Granito, A.; Ponti, F.D.; Corinaldesi, R.; Bianchi, F. Clinical findings and anti-neuronal antibodies in coeliac disease with neurological disorders. Scandinavian journal of gastroenterology 2002, 37, 1276-1281.
